# All-synchronized picosecond pulses and time-gated detection improve the spatial resolution of two-photon STED microscopy in brain tissue imaging

Hirokazu Ishii [1,2,3], Kohei Otomo[1,2,4], Ching-Pu Chang[1,2,5], Miwako Yamasaki[5], Masahiko Watanabe[5], Hiroyuki Yokoyama[6], Tomomi Nemoto[1,2,3]*

**1** Exploratory Research Center on Life and Living Systems (ExCELLS), National Institutes of Natural Sciences, Okazaki, Japan, **2** National Institute for Physiological Sciences (NIPS), National Institutes of Natural Sciences, Okazaki, Japan, **3** School of Life Science, The Graduate University for Advanced Studies (SOKENDAI), Okazaki, Japan, **4** Graduate School of Medicine, Juntendo University, Tokyo, Japan, **5** Faculty of Medicine, Hokkaido University, Sapporo, Japan, **6** New Industry Creation Hatchery Center (NICHe), Tohoku University, Sendai, Japan

* tn@nips.ac.jp

**Data Availability Statement:** All relevant data are within the paper and its Supporting information files.

## Abstract

Super-resolution in two-photon excitation (2PE) microscopy offers new approaches for visualizing the deep inside the brain functions at the nanoscale. In this study, we developed a novel 2PE stimulated-emission-depletion (STED) microscope with all-synchronized picosecond pulse light sources and time-gated fluorescence detection, namely, all-pulsed 2PE-gSTED microscopy. The implementation of time-gating is critical to excluding undesirable signals derived from brain tissues. Even in a case using subnanosecond pulses for STED, the impact of time-gating was not negligible; the spatial resolution in the image of the brain tissue was improved by approximately 1.4 times compared with non time-gated image. This finding demonstrates that time-gating is more useful than previously thought for improving spatial resolution in brain tissue imaging. This microscopy will facilitate deeper super-resolution observation of the fine structure of neuronal dendritic spines and the intracellular dynamics in brain tissue.

## Introduction

Two-photon excitation (2PE) microscopy is widely used in neuroscience research [1]. The use of near-infrared pulsed laser beams for 2PE offers superior penetration depths and low invasiveness for biological specimens [2, 3]. However, the visualization of nanoscale phenomena, such as morphological changes in neuronal dendritic spines associated with synaptic plasticity, requires super-resolution technologies. One such technology is stimulated-emission-depletion (STED) microscopy, where a doughnut-shaped beam induces stimulated emission and restricts the area of natural fluorescence emission below the diffraction limit of the excitation laser beam [4].

Over the past decade, the spatial resolution of 2PE microscopy has been improved by combining STED technologies [5–12]. This improvement promises new approaches to visualizing

**Funding:** This study was supported by: • the AMED Brain/MINDS (JP19dm0207078) • the JST CREST (JPMJCR20E4) • the MEXT/JSPS KAKENHI (JP15H05953 "Resonance Bio,"), (JP16H06280 "Advanced Bioimaging Support,"), (JP18K14659), (JP20H00523), (JP20H05669), (JP21K19346), (JP22H02756) • the Research Foundation for Opto-Science and Technology • the ExCELLS "Encouragement Research for Young Scientists". The funders had no role in study design, data collection and analysis, decision to publish, or preparation of the manuscript.

**Competing interests:** The authors have declared that no competing interests exist.

the morphology and function of cells such as neurons, but these approaches are not widely adopted. One possible explanation is their complex systems; an ultrashort-pulsed near-infrared light source is used for 2PE, and an additional optical path with a wave-front modulator functions as the STED light source. A light source system combining two commercial mode-locked Ti:sapphire laser light sources were used in previous studies; one Ti:sapphire laser is for 2PE, and the other is for STED, where an optical parametric oscillator is used to obtain the desired wavelength for synchronization with the 2PE pulse [7, 10]. The single-wavelength approach, where one mode-locked Ti:sapphire laser light source is used for both 2PE and STED, simplifies the optical setup, but the available probes are limited [13]. An easy-to-use 2PE-STED microscopy method is required to accelerate the nanoscale visualization of brain functions in many laboratories.

We previously developed a pulsed 2PE STED (2PE-STED) microscopy technique that incorporates electrically controllable components for easy, effective imaging [14–16]. One of the key components is a set of transmissive liquid-crystal devices (tLCDs), which converts the STED beam into the shape of a doughnut and adjusts the focal position along the optical axis. Unlike conventional reflective spatial light modulators (SLMs), tLCDs are installed directly before the objective lens (that is, without adding an optical path), thus allowing for a compact setup. A semiconductor-laser–based pulsed light source system for 2PE and STED is another key component [17, 18]. Utilizing pulsed laser light for STED reduces the required average power and thus causes less photodamage to specimens than continuous-wave laser light sources. The combination of these components in 2PE-STED microscopy results in a spatial resolution that is approximately five times higher than that of 2PE microscopy without severe photobleaching [16]. However, the practical use of 2PE-STED microscopy is hindered by spatial resolution degradation caused by background signals from residual fluorescence due to incomplete STED, optical scattering, and the autofluorescence of biomaterials due to the direct excitation of the STED beam.

Time-gated fluorescence detection is used in STED microscopy to improve the spatial resolution by reducing signals caused by incomplete depletion [19]. However, the impact of time-gating has been demonstrated theoretically and experimentally to be negligible when the pulse width of the STED beam is shorter than half the fluorescence lifetime of the target probe [20, 21]. Time-gating is also useful for removing background signals caused by optical scattering and biomaterial autofluorescence, even in the case of STED beams with short pulse widths [20, 21]. This advantage might be critical for improving the spatial resolution of thick-sample imaging, where background signals are more severe than those in thin-sample imaging, but the effect has not been estimated well [9].

In this study, we developed a novel 2PE-STED microscope with all-synchronized picosecond pulse light sources and time-gated detection, namely, the all-pulsed 2PE-gSTED microscope. This system is based on our previous 2PE-STED microscope, whose advantages we maintained while successfully reducing undesirable signals via time-gating. The spatial resolution of the image of a fixed brain slice was better by approximately 1.4 times compared with that of a non-time-gated image. Therefore, the time-gating system was critical to dissociating the undesirable signals in brain tissue imaging.

## Results

The time-gated detection was implemented in the 2PE-STED microscope we previously developed [16] using a time-correlated single-photon counting (TCSPC) module. We termed this system as an all-pulsed 2PE-gSTED microscope. To our knowledge, this is the first example of a gated STED system that uses pulsed light sources for both 2PE and STED. The time delays of

both optical pulses generated from the laser-diode (LD)-based light source system and the TCSPC module were required to be tightly controlled with picosecond precision. Actually, the synchronization could be easily achieved using only one electrical timing controller because it was based on an electrically controllable LD-based light source system. The optical setup was kept compact (Fig 1) using the tLCDs as a SLMs. First, we compared the spatial resolution improvements of our 2PE-STED microscopy for visualizations of coverslip-embedded fluorescent beads with and without time-gated detection. The fluorescence images of 20-nm Nile red beads were obtained by overlaying a 1064-nm 2PE beam with a 655-nm STED beam shaped into a doughnut using tLCDs. In addition, using the tLCDs, the axial focal position of the 655-nm STED beam was matched with that of the 2PE beam, and the STED beam was converted to be circularly polarized. The average power of the 655-nm STED and 2PE beams at the focal plane were both 3.0 mW, respectively. The fluorescent images were fitted with a Gaussian function and the full width at half maximum (FWHM) values along the x-axis and y-

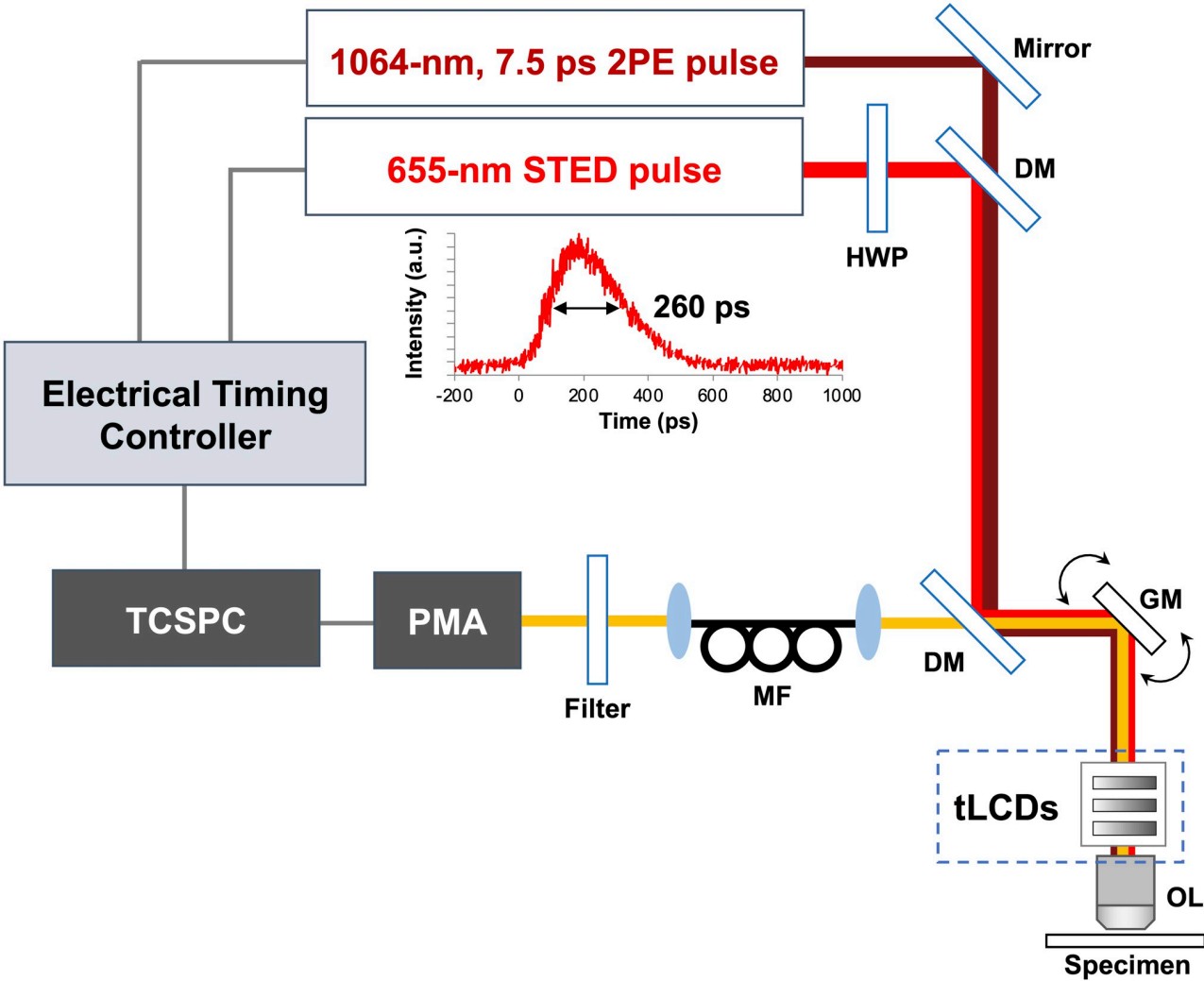

**Fig 1. Optical setup for all-pulsed 2PE-gSTED microscopy.** DM: dichroic mirror, GM: galvanometer mirrors, MF: multimode fiber, PMA: hybrid photomultiplier detector assembly, HWP: half-wave plate, TCSPC: time-correlated single-photon counting, tLCDs: transmissive liquid-crystal devices, OL: objective lens. The optical waveform of the 655-nm STED pulse is shown in the middle. STED pulse width = 260 ps, 2PE pulse width = 7.5 ps.

axis were calculated. Without time-gating, the FWHM values in the 2PE-STED image reached 97.2 ± 1.6 nm and 100.2 ± 3.0 nm along the x-axis and y-axis, respectively, by applying the maximum average power of the STED beam (Fig 2). Because the 2PE beam was elliptically polarized using the tLCDs (Materials and methods), the focal spot was elongated along the polarization direction (= the y-axis in our setup). Then, we searched for the gate window $\Delta T$ (= $T_{finish}$ − $T_{initial}$) that gave the best FWHM values in the bead images (S1 Fig). Vicidomini et al. [21] theoretically revealed that the sharpest effective point spread function is obtained when photons are collected immediately after STED in pulsed STED microscopy. Late detection before the next excitation pulse will lead to the dominance of background signals. Thus, we set $T_{initial}$ to 0.25 ns or 0.50 ns (time-bin width = 0.25 ns, STED pulse width = 260 ps) and $T_{finish}$ was set arbitrarily (S1 Fig). Setting $\Delta T$ to 0.50–7.00 ns gave the FWHM values, which were 88.4 ± 1.5 nm and 91.7 ± 2.7 nm along the x-axis and y-axis, respectively (Fig 2). They were improved by a factor of 1.1 compared to the 2PE-STED images without time-gating. Even when the STED light power was changed to 1.0 mW, in which incomplete signal was thought to be relatively increased, the spatial resolution was improved similarly by 1.1 times with time-gating. The effectiveness of time-gating was also assessed by comparing the peak

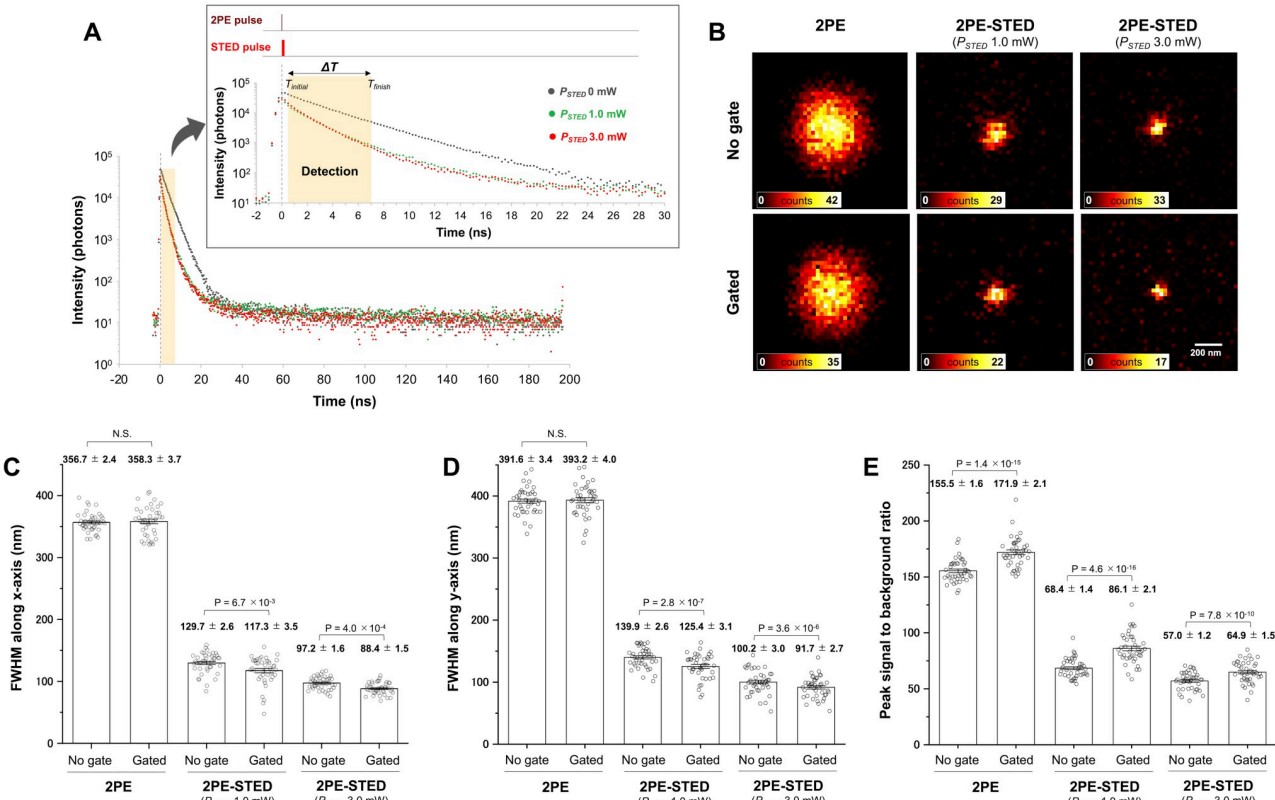

**Fig 2. Improving spatial resolution of the bead images with time-gating.** (A) Histogram of the photon-arrival times of the fluorescence of the 20-nm Nile red beads in 2PE or 2PE-STED microscopy. The time course of two-photon excitation (2PE), stimulated-emission-depletion (STED) and fluorescence signal detection are shown at the top of the box. The time-gated detection is characterized by $\Delta T$ (= $T_{finish}$ − $T_{initial}$). Time-bin width = 250 ps, 2PE pulse width = 7.5 ps, STED pulse width = 260 ps. (B) Comparison of 2PE and 2PE-STED imaging of 20-nm Nile red bead with or without time-gating ($T_{initial}$ = 0.50 ns, $T_{finish}$ = 7.00 ns, $\Delta T$ = 6.50 ns). The FWHM values along the x-axis (C) and y-axis (D) were analyzed and compared from 41 images of a single Nile red bead. (E) The peak signal-to-background ratio (PSBR) of each image was obtained by dividing the peak intensity by the mean background intensity (Materials and methods). Error bars represent the standard error of the mean (s.e.m.). P-values are from the paired Student's t-test.

signal-to-background ratios (PSBRs) of the time-gated and non-time-gated images (Fig 2E). Time-gating decreased the fluorescence peak and mean background intensities (S2 Fig) and increased the PSBRs of all images, including the 2PE images.

We used all-pulsed 2PE-gSTED microscopy to observe the fine structure of neuronal dendritic spines in thick brain tissue as a proof of concept in a biological application. We perfusion-fixed Thy1-YFP-H mice brains, expressing the enhanced yellow fluorescent protein (EYFP) in a certain percentage of the neuronal population, and sliced the brain tissue into 200-μm-thick slices. The slices were treated with a Sca*l*eA2 solution, which can hyperhydrate and delipidate the mouse brains to clear the tissues and minimize optical scattering and aberration, thus improving image quality [22]. We focused on the neuronal dendrites located approximately 4 μm and 36 μm from the surface of the coverslip in the 2PE-gSTED microscope (Fig 3). The average power of the 655 nm STED and 2PE beams at the focal plane was 3.0 mW and 3.3 mW, respectively. The FWHM values along the spine neck at the 4 μm and 36 μm depths in the 2PE-STED images were 218.6 ± 9.1 nm and 231.5 ± 9.0 nm, respectively. Then, the FWHM was evaluated at $T_{initial}$ values of 0.25 ns, 0.50 ns, and 1.25 ns and arbitrary $T_{finish}$ values. Unlike in the case of the bead images (S1 Fig), we also evaluated the FWHM at $T_{initial}$ = 1.25 ns because we expected the autofluorescence inherent in biomaterials, which has a shorter lifetime than that of the target fluorophore (S3 Fig). Theoretical results have

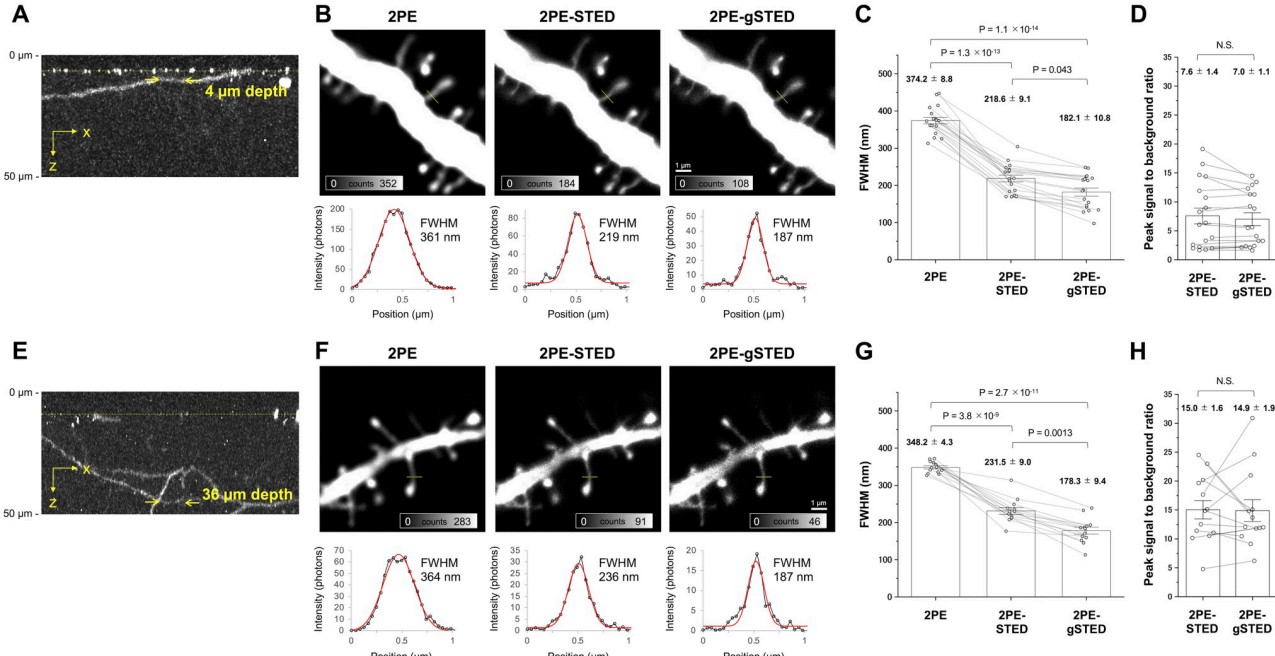

**Fig 3. Spatial resolution of the fixed brain slice images with or without time-gating.** (A, E) 2PE images of neuronal dendrites in the fixed brain slice, reconstructed from z-stacks. The x–z images show the maximum intensity projections. The dashed yellow lines indicate the coverslip surfaces marked with 250 nm Nile red beads. (B, F) Comparison of 2PE, 2PE-STED, and gated 2PE-STED (2PE-gSTED) images of the neuronal dendrites. (B) The imaging depth was approximately 4 μm from the surface (arrows in A). The gate window was set to $T_{initial}$ = 1.25 ns, $T_{finish}$ = 14.00 ns, and $\Delta T$ = 12.75 ns. (F) The imaging depth was approximately 36 μm from the surface (arrows in E). The gate window was set to $T_{initial}$ = 1.25 ns, $T_{finish}$ = 7.00 ns, and $\Delta T$ = 5.75 ns. The graphs show the fluorescence intensity profiles across the spine necks in B and E (yellow lines). The red lines indicate the fitted Gaussian curves of the measurement values (black dots). The FWHM values of the Gaussian curves were calculated using the width parameters (Materials and methods). (C, G) The FWHM values were compared between the fluorescence intensity profiles across the spine necks at the depths of 4 μm (N = 18, including the value from the line in B) and 36 μm (N = 13, including the value from the line in F). (D, H) The PSBR of each Gaussian curve in the 2PE-STED and 2PE-gSTED images was obtained by dividing the amplitude by the offset. The error bars represent the s.e.m. The P-values are from the Welch's t-test with Bonferroni correction in C and G and from the paired Student's t-test in D and H.

demonstrated that the use of an initial time ($T_{initial}$) larger than the STED pulse width reduces the brightness without further reducing the FWHM, and we experimentally evaluated the time-gating effects on brain tissues for the first time. The best FWHM values were obtained at $T_{initial}$ = 1.25 ns in the images at the 4 μm and 36 μm depths. Setting $\Delta T$ to 1.25–14.00 ns for the 2PE-STED image at the 4 μm depth gave an FWHM value of 182.1 ± 10.8 nm, whereas setting $\Delta T$ to 1.25–7.00 ns for the 2PE-STED image at the 36 μm depth gave an FWHM value of 178.3 ± 9.4 nm. These FWHM values were inferior to those in the 2PE-STED images of the Nile red beads (Fig 2). Both the fluorescence peaks and background intensities were decreased by time-gating (S4 Fig), but the PSBRs did not change significantly.

The wavelength of the STED beam was 655 nm, which was not ideal for STED imaging of EYFP because the fluorescence depletion efficiency for EYFP is approximately 70%, which is relatively low compared with approximately 95% of Nile red [16]. Therefore, we labeled EYFP expressed in the neurons with AlexaFluor532 by immunostaining. The 655-nm STED beam showed relatively high depletion efficiencies of approximately 90% against AlexaFluor532 [16]. Each fixed brain slice was cleared with Sca*l*eA2 [22] and immunostained with anti-GFP, followed by AlexaFluor532-conjugated second antibodies. Then, we focused on the neuronal dendrites located approximately 7 μm from the surface of the coverslip. At this depth, the antibodies were able to sufficiently penetrate and label the EYFP expressed in the neurons. The average power of the 655-nm STED and 2PE beams at the focal plane was 3.0 mW. The FWHM value along the spine neck was evaluated with the gate window (S5 Fig). Setting $\Delta T$ to

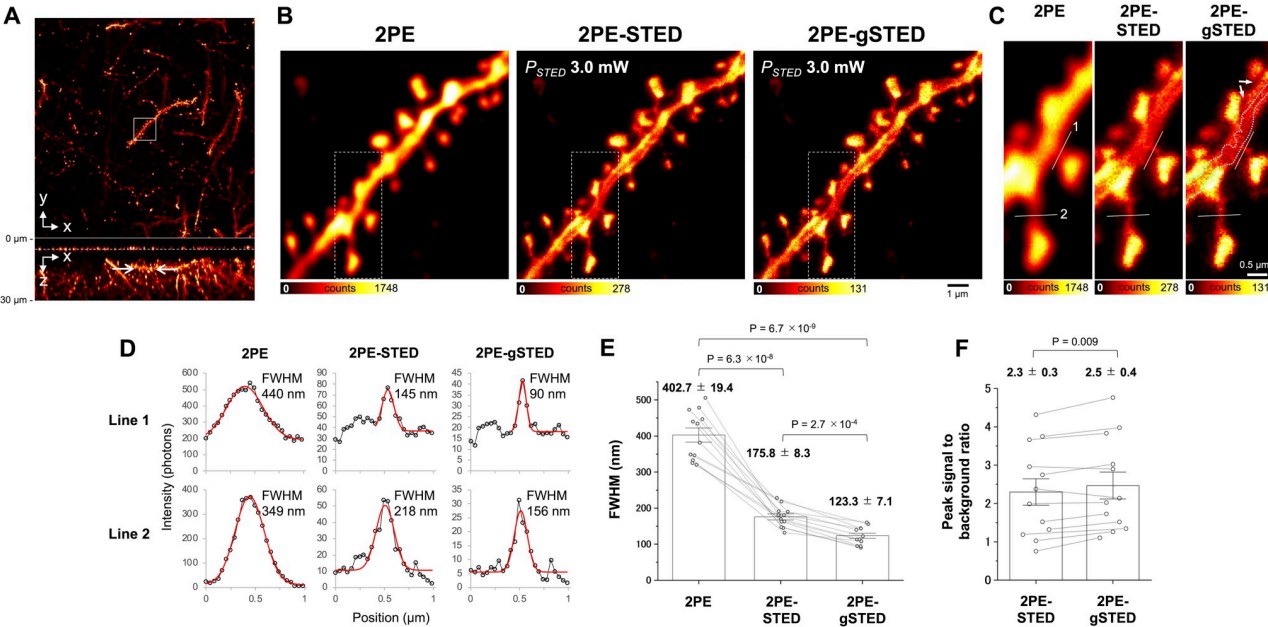

**Fig 4. Improving spatial resolution of the immunostained fixed brain slice with time-gating.** (A) 2PE image of neuronal dendrite in fixed brain slice immunostained with anti-GFP followed by AlexaFluor532-conjugated secondary antibodies, reconstructed from z-stacks. The x-y and x-z images show max intensity projections. (B) Comparison of 2PE 2PE-STED, and gated 2PE-STED (2PE-gSTED; $T_{initial}$ = 0.50 ns, $T_{finish}$ = 3.50 ns, $\Delta T$ = 3.00 ns) images of the area marked with a white square in A. The imaging depth was approximately 7 μm (arrows in A) from the coverslip surface marked with the 250-nm Nile red beads (dashed line). (C) Enlarged view of each area marked with square dashed lines in B. (D) Fluorescence intensity profiles across the spine necks indicated with lines 1 and 2 in C. The red lines indicate the fitted Gaussian curves of the measurement values (black dots). The FWHM values of the Gaussian curves were calculated using the width parameters (Materials and methods). (E) The FWHM values were compared between the fluorescence intensity profiles across the spine necks (N = 12, including the values from lines 1 and 2). (F) The PSBR of each Gaussian curve in the 2PE-STED and 2PE-gSTED images was obtained by dividing the amplitude by the offset. The error bars represent the s.e.m. The P-values are from the Welch's t-test with Bonferroni correction in E and from the paired Student's t-test in F.

0.50–3.50 ns gave the best FWHM. Compared to the 2PE image, the fine structure and morphological individualities of each spine head were visualized in the 2PE-gSTED image. The FWHM value along the spine neck in the 2PE-gSTED image was 123.3 ± 7.1 nm, which was approximately 1.4 times higher than that in the image without time-gating (2PE-STED). The line 1 profile in the 2PE image seemingly contained two peaks that blended into one (Fig 4D), which could explain the outliers in Fig 4E. The peak was resolved into two peaks in the 2PE-gSTED image, and one of them reached an FWHM value of 90 nm (Fig 4C). Although the fluorescence peak and background intensities were decreased by time-gating (S6 Fig), the PSBR in the gated 2PE-STED image was larger than that in the non-gated image (Fig 4F). This distinguished the boundaries between the spine and the shaft (arrows). Moreover, the 2PE-STED microscope visualized the subcellular structures inside the dendrite shafts as shadows (dotted line).

## Discussion

For the first time, we developed a gated STED system that uses pulsed light sources for both 2PE and STED, namely, all-pulsed 2PE-gSTED microscopy (Fig 1). The spatial resolutions were improved by 1.1 times for the fluorescent beads (Fig 2) compared with 2PE-STED. The arriving times of fluorescent photons are spatially distributed at the focal plane by overlapping the doughnut-shaped STED beam; the earliest arriving photons are from fluorophores located in the regions with the highest STED beam intensity [19, 20]. Therefore, the implementation of the time-gated detection system was achieved by discarding early arriving photons due to incomplete depletion signal by STED, which negatively affects the spatial resolution.

The 2PE-gSTED microscope visualized the subcellular structures inside the dendrite shafts as shadows (Fig 4). These shadows may have been derived from intracellular organelles, including the endoplasmic reticulum network and mitochondria, because the EYFP labeled with AlexaFluor532 was localized in the cytosol. Such shadow imaging has been proposed as a super-resolution shadow imaging technique [23] based on 3D-STED microscopy. This microscopy approach can improve the spatial resolution in both the planar and axial directions by surrounding the dark spot of the STED beam on the excitation focal spot [23]. The spatially distributed organelles inside the neuronal dendrites might be visualized more precisely by upgrading the 2PE-gSTED microscope to a 3D-STED one. However, the acquisition time of the 2PE-gSTED image of the brain slice immunostained with AlexaFluor532 (Fig 4B) was 1 min under the current experimental condition. We could lengthen the acquisition time to collect enough photons to construct an image for fixed samples because the 2PE-gSTED microscope had no severe photobleaching; however, this acquisition speed was too slow to visualize the cellular dynamics of living cells or tissues. Photobleaching could not be eliminated completely. This was evident in bead imaging, where the total number of photons in the 2PE-STED image at $P_{STED}$ = 1.0 mW was almost the same as that in the 2PE-STED image at $P_{STED}$ = 3.0 mW, which was captured first (Fig 2A). This result was probably caused by photobleaching, as the total number of photons should decrease as the STED power increases. The screening of fluorescent probes with the highest possible brightness at the 2PE wavelength and high STED efficiency is important for accelerating the imaging of biological dynamics. Narrowing the field of view can also increase the imaging speed, but the fundamental solution is to increase the repetition rates of the light sources of the 2PE-gSTED microscope. The repetition rate was 5 MHz, which was slower than those in previous studies, which were mainly 40–80 MHz [9, 19, 21]. The repetition rate of our light sources can be increased, but doing so will reduce the pulse energy. Thus, the STED pulses of the current system cannot provide efficient

depletion at a higher repetition rate. The LD-based light sources should be upgraded such that the repetition rate increases but the optical properties are retained, including the peak power, to maintain 2PE and STED efficiencies for imaging.

The spatial resolution of the immunostained brain slice was drastically improved (1.4 times), which was greater than that for the fluorescent beads (1.1 times) (Fig 4). This effectiveness, particularly for biological specimens, may be due to the removal of the background signals such as optical scattering from the brain tissues and autofluorescence inherent in biomaterials, which have a shorter or longer fluorescence lifetime than the target fluorophore. For example, flavin adenine dinucleotide, a major biomaterial emitting autofluorescence, has an absorption in the visible range and emits fluorescence centered on 520 nm. It has a short fluorescence lifetime ranging at approximately 0.04–0.13 ns in the protein-bound state [24] and can be removed by time-gating with the window setting ($T_{initial}$ = 0.50 ns). Generally, autofluorescence emission from the sample is almost unavoidable for fluorescence microscopy, including 2PE and STED microscopy. The optical scattering and autofluorescence excited using the STED beam must be more severe for thicker biological samples such as brain tissue because the STED intensity is distributed three-dimensionally. In fact, the spatial resolution of the spine neck at the 36 μm depth was improved by 1.3 times by time-gating. This improvement rate was larger than that at the 4 μm depth (Fig 3). The background signal relative to the fluorescence signal was higher in the 2PE-STED image at the 36 μm depth than at the 4 μm depth (S3A and S3D Fig). Hence, the effect of scattering and autofluorescence may have increased according to the depth of the imaging area.

Previous studies using gated STED microscopy mainly targeted relatively thin biological samples, such as cultured cells; the beneficial point of time-gating for thicker samples might be overlooked [9]. In addition, this supposition is consistent with the results for the fluorescent bead (Fig 2B) and the theoretical demonstration of Vicidomini et al. [21]. Simple time-gating is insufficient for improving the spatial resolution of pulsed STED microscopy in the case of subnanosecond STED pulses. Therefore, we successfully demonstrated that the time-gating system was critical to dissociating the targeted fluorescence signals caused by optical scattering and autofluorescence and improved the spatial resolution in the thick biological sample. However, time-gating also rejects photons from the center of the focal spot, resulting in brightness reduction, which might suppress the abovementioned resolution improvement. This problem has been solved through the separation of photons by lifetime tuning (SPLIT) method [25]. This analysis approach uses phasor analysis to efficiently distinguish photons emitted from the center and the periphery of the excitation spot. The spatial resolution of brain tissue imaging using our microscope may be enhanced further via SPLIT analsis.

In conclusion, this study demonstrated that time-gating is more useful for improving spatial resolution in thick brain tissue. Combined with the advantage of the time-gating or other photon separation analyses such as SPLIT, all-pulsed 2PE-gSTED microscopy is expected to facilitate a deeper super-resolution observation to shed light on the brain functions at the nanoscale.

## Materials and methods

### All-pulsed 2PE-gSTED microscope

As shown in Fig 1, the optical setup was based on that of our previous 2PE-STED microscope [16]. The setup included two types of pulsed LD-based light sources driven by a custom-built electrical pulse generator at a repetition rate of 5 MHz. For 2PE, a 7.5 ps optical pulse source consisting of an in-house 1064 nm gain-switched LD and optical fiber amplifier chain was employed [17, 26]. Using this laser system, we visualized the structure of the hippocampal

dentate gyrus neurons expressing EYFP [26] and the neuronal activity in the hippocampal CA1 region without resectioning the neocortex in the anesthetized mice brains [27]. While the pulse propagated through the optical materials, its width was maintained at 7.5 ps. Unlike a femtosecond light source, the pulsed LD-based light source did not require a negative chirp system. For STED, the optical pulse source was constructed using a 1.3 μm gain-switched semiconductor-laser optical amplifier under continuous-wave laser light injection. The seed pulses were amplified using a praseodymium-doped fiber amplifier and converted into second-harmonic pulses with a pulse width of 260 ps, a peak wavelength of 655 nm, and an average power of 7 mW. The laser beams were directed through a single optical pass using a dichroic mirror (FF775-Di01; Semrock, NY, US), and the relative delay timing between the 2PE and STED pulses was measured using a photodetector (1414; New Focus, CA, US) and a sampling oscilloscope (TDS8200, Tektronix, OR, US). Then, an electrical timing controller (T560; Highland Technology, CA, US) was used to add delay to the 2PE pulse at a 10 ps resolution to make it overlap the initial rise point of the STED pulse. In a past study, we confirmed that the relative position of each pulse gave the best STED efficiency [16]. Once the setting of the T560 was determined, the pulses could be reproducibly synchronized when the light sources were turned on. The pulse synchronization did not have to be checked using the oscilloscope each time. The synchronized pulsed beams were introduced into an upright microscope (ECLIPSE FN1; Nikon, Tokyo, Japan) equipped with a galvano-mirror scanner (C2; Nikon). The tLCDs were inserted between the microscope and a water-immersion objective lens (CFI Plan Apo IR 60XWI, Nikon; 1.27 numerical aperture). Further information about tLCDs can be found in the literature [15]. The set of tLCDs in the current work was composed of three different tLCDs; the optical properties were modified by altering the voltages applied to the liquid-crystal molecules. The first tLCD was a gradient-index lens based on a tLCD; it could control the focal spot of the STED beam along the z axis to compensate for the chromatic aberration between the STED and 2PE beams. The second tLCD converted the STED beam into an optical vortex. These tLCDs did not affect the 2PE beam, which was polarized orthogonal to the orientation of the liquid-crystal molecules. The third tLCD was a plain-cell tLCD; it functioned as a variable wave plate to convert the STED beams into circular polarization. The polarization of the 2PE beam was also modulated to an elliptical state using the plain-cell tLCD.

The fluorescence was descanned and focused onto a multimode optical fiber connected to a hybrid photomultiplier detector assembly (PMA Hybrid; PicoQuant, Berlin, Germany). The fiber provided detection optics confocality to reduce out-of-focus fluorescence [23], and the output signal was cleaned using a band-pass filter (FF01-596/83-25; Semrock) placed before the detector. For EYFP imaging, the band-pass filter was replaced with a D535/50M (Chroma, VT, US). The detector signals were acquired using a TCSPC module (TimeHarp 260; PicoQuant) synchronized with a trigger signal provided by the electrical timing controller (T560); the photons were counted, and their arrival time was measured for time-gating. The imaging samples were placed on a custom-built microscope stage equipped with a horizontal adjustment mechanism. The stage angle was manually controlled to minimize the aberration generation caused by the tilt of the coverslip on the sample [28].

## Fluorescent beads

Nile red–labeled beads (20 nm diameter; Invitrogen, MA, US) were diluted with distilled water (1:2,500–1:5,000, v/v). The bead solutions were added dropwise to high-tolerance coverslips (0.17 ± 0.005 mm thickness; Matsunami, Osaka, Japan), followed by drying and mounting using a ProLong diamond reagent (Invitrogen). Fluorescent images of the beads were obtained at a pixel size and dwell time of 28 nm × 28 nm and 40 μs, respectively.

## Animal

A Thy1-YFP-H mouse line [29] was housed at a temperature of 22˚C–24˚C under a standard 12 h light–dark cycle with ad libitum access to water and standard chow. All animal studies were performed on the basis of the Animal Research: Reporting of In Vivo Experiments guidelines, approved by the Institutional Animal Care and Use Committee of the National Institute of Natural Sciences, and conducted according to the guidelines of the National Institute for Physiological Sciences (Approval Nos. 20A017 and 21A046).

## Preparation of fixed brain slices

Thy1-YFP-H mice were anesthetized with isoflurane and perfused with saline, followed by 4% paraformaldehyde (PFA, in 0.1 M PB). The brain was surgically removed and treated with 4% PFA (in 0.1 M PB) at 4˚C overnight, followed by the replacement of 4% PFA with 0.1 M PB. 200-μm coronal brain slices were prepared using a vibratome (7000smz, Campden Instruments, Leicestershire, UK) and treated with a Sca*l*eA2 solution (4 M urea, 10% [w/v] glycerol, and 0.1% [w/v] Triton X-100) at 37˚C for 2 days, as previously reported [22]. The EYFP expressed in the neurons was labeled as follows: the specimens were immunostained with an anti-GFP antibody (1 μg/ ml; RRID AB_2571573) [30] in phosphate-buffered saline with 0.1% Triton X-100, followed by anti-rabbit antibodies conjugated with AlexaFluor532 (1:200 dilution; A11009, Molecular Probes). The cleared or immunostained specimens were placed on the high-tolerance coverslip with 250 nm Nile red beads and mounted with Sca*l*eA2 in the same way. The beads were used as an indicator to visualize and control the tilt of the coverslip on the microscope stage. The fluorescence images had a pixel size and dwell time of 41 nm by 41 nm and 20 μs, respectively. The image acquisition times were 5 min for the cleared specimens and 1 min for the immunostained specimens. Low-magnification 2PE images were obtained using a gallium arsenide phosphide–based non-descanned detector in the 2PE-gSTED microscope. For the imaging of the cleared specimens, 50-μm-thick z-stacks were acquired at 0.50 μm intervals at a pixel size of 414 nm × 414 nm and a pixel dwell time of 4 μs (Fig 3A) or a pixel size of 207 nm × 207 nm and a pixel dwell time of 10 μs (Fig 3E). For the imaging of the immunostained specimens, 30-μm-thick z-stacks were acquired at 0.20 μm intervals at a pixel size of 207 nm × 207 nm and a pixel dwell time of 10 μs (Fig 4A).

## Image analysis

All fluorescence intensity profiles were obtained using single-pixel-wide lines in the software Fiji. For the beads, the fluorescence intensity profiles across the central intensity were obtained along the x and y axes. For the spine necks, the lines were placed across the spine necks manually. The line profiles were fitted with a Gaussian function (GaussAmp in the software Origin) to find the parameters of the amplitude $A$, offset $y0$, width $w$, and center $x_c$ as follows:

$$y = y_0 + Ae^{-\frac{(x-x_c)^2}{2w^2}}.$$

The FWHM of the Gaussian function was calculated using the width parameter:

$$\text{FWHM} = 2w\sqrt{\ln(4)}.$$

We used Fiji to measure the peak intensity and mean background intensity of each bead image. The region of interest (ROI) was manually drawn outside the bead as the background region in the 2PE image. The same ROI was placed on all bead images, and each mean intensity was measured as the mean background intensity. The fluorescence peak intensities were calculated by subtracting the background intensities from the raw peak intensities. Then, the

PSBR was obtained by dividing the fluorescence peak intensity by the mean background intensity. We used the parameters obtained from the Gaussian curves to calculate the PSBR in the local area around the spine neck. The amplitude and offset were defined as the fluorescence peak intensity and background intensity, respectively. The PSBR for the spine neck was obtained by dividing the amplitude by the offset.

Origin was used for statistical analysis. The data in the text and figures are the mean ± standard error of the mean (s.e.m.).

## Supporting information

**S1 Fig. Dependence of full width at half maximum (FWHM) values on gate windows.** The FWHM values along the x-axis were evaluated within each gating window for the final image of all 41 images of a single Nile red bead from 2PE-STED imaging (analyzed in Fig 2). The dotted red line indicates the FWHM value of the final 2PE-STED image without time-gating (no gate). $T_{initial}$ was set to 0.25 ns or 0.50 ns, and $T_{finish}$ was set arbitrarily. Setting $\Delta T$ to 0.50–7.00 ns gave the best FWHM (asterisk).
(TIF)

**S2 Fig. Fluorescence peak and mean background intensities of the bead images.** (A) Comparison of the 2PE, 2PE-STED, and gated 2PE-STED (2PE-gSTED) imaging of the 20 nm Nile red bead used in Fig 2. Fluorescence peak intensities of all 41 images of a single Nile red bead under each condition. The error bars represent the s.e.m. The P-values are from a paired Student's t-test. (B) The same ROI was manually set to the background region of the 2PE, 2PE-STED, and 2PE-gSTED images, and the mean intensities in the ROIs were compared under each condition. The error bars represent the s.e.m. The P-values are from a paired Student's t-test.
(TIF)

**S3 Fig. Gate window for the fixed brain slice at each imaging depth.** (A, D) Histogram of the photon-arrival times of the fluorescence of the neuronal dendrite observed with 2PE or 2PE-STED microscopy at 4 μm depth (A) and 36 μm depth (D). The constructed images from the data in A and D were shown in Fig 3B and 3F, respectively. The photon-arrival times from the background region of the 2PE-STED images at the 4 μm and 36 μm depths (B and E, respectively) are also plotted on each histogram. (C, F) The FWHM values along the spine necks in the 2PE-STED images at both depths were evaluated within each gating window. The dotted red line indicates the FWHM value without time-gating (no gate). $T_{initial}$ was set to 0.25 ns, 0.50 ns, and 1.25 ns. $T_{finish}$ was set arbitrarily. The best FWHM value was obtained (asterisks) at $\Delta T$ = 1.25–14.00 ns for the 4-μm-depth image and $\Delta T$ = 1.25–7.00 ns for the 36-μm-depth image.
(TIF)

**S4 Fig. Fluorescence peak and background intensities of the fixed brain slice.** The fluorescence peak and background intensities in the local area around the spine necks in the 2PE-STED and 2PE-gSTED images used in Fig 3 were obtained from the Gaussian curve parameters. The fluorescence peak intensities were obtained from the amplitude parameters, and the background intensities were obtained from the offset parameters at the depths of 4 μm (A) and 36 μm (B). The error bars represent the s.e.m. The P-values are from a paired Student's t-test.
(TIF)

**S5 Fig. Gate window for the immunostained fixed brain slice.** (A) Histogram of the photon-arrival times of the fluorescence of the neuronal dendrites in the immunostained fixed brain

slices observed via 2PE or 2PE-STED microscopy. The constructed images are shown in Fig 4. (B) The FWHM values along the spine neck in the 2PE-STED image were evaluated within each gating window. The dotted red line indicates the FWHM value without time-gating (no gate). $T_{initial}$ was set to 0.25 ns, 0.50 ns, and 1.25 ns. $T_{finish}$ was set arbitrarily. Setting $\Delta T$ to 0.50–3.50 ns gave the best FWHM (asterisk).
(TIF)

**S6 Fig. Fluorescence peak and background intensities of the immunostained fixed brain slice.** The fluorescence peak and background intensities in the local area around the spine necks in the 2PE-STED and 2PE-gSTED images in Fig 4 were obtained from the Gaussian curve parameters. The fluorescence peak intensities were obtained from the amplitude parameters, and the background intensities were obtained from the offset parameters. The error bars represent the s.e.m. The P-values are from a paired Student's t-test.
(TIF)

**S1 Raw images. All raw images used in Figs 2 to 4.**
(ZIP)

# Acknowledgments

We thank Dr. S. Sato and Dr. Y. Kozawa of Institute of Multidisciplinary Research for Advanced Materials, Tohoku University for their helpful advice regarding optical setup. We also thank Dr. N. Hashimoto, Mr. M. Kurihara and Dr. A. Tanabe of Citizen Watch Co., Ltd. for kindly providing the tLCDs. We are grateful for the technical support and advice provided by Dr. R. Enoki, Dr. M. Tsutsumi and Dr. J. Sakamoto of Exploratory Research Center on Life and Living Systems (ExCELLS) and National Institute for Physiological Sciences (NIPS), National Institutes of Natural Sciences.

# Author Contributions

**Conceptualization:** Hirokazu Ishii, Kohei Otomo, Hiroyuki Yokoyama, Tomomi Nemoto.

**Data curation:** Hirokazu Ishii, Kohei Otomo, Ching-Pu Chang, Tomomi Nemoto.

**Formal analysis:** Hirokazu Ishii, Kohei Otomo, Tomomi Nemoto.

**Funding acquisition:** Hirokazu Ishii, Kohei Otomo, Tomomi Nemoto.

**Investigation:** Hirokazu Ishii, Ching-Pu Chang.

**Methodology:** Hirokazu Ishii, Ching-Pu Chang, Miwako Yamasaki, Masahiko Watanabe, Hiroyuki Yokoyama, Tomomi Nemoto.

**Project administration:** Tomomi Nemoto.

**Resources:** Ching-Pu Chang, Miwako Yamasaki, Masahiko Watanabe.

**Supervision:** Tomomi Nemoto.

**Validation:** Hirokazu Ishii, Ching-Pu Chang.

**Visualization:** Hirokazu Ishii.

**Writing – original draft:** Hirokazu Ishii, Kohei Otomo, Tomomi Nemoto.

**Writing – review & editing:** Hirokazu Ishii, Kohei Otomo, Ching-Pu Chang, Miwako Yamasaki, Masahiko Watanabe, Hiroyuki Yokoyama, Tomomi Nemoto.

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
