## [Decision Letter · Decision Letter 0]

29 May 2023

PONE-D-23-09651All-electrically synchronized picosecond pulses and time-gated detection improve the spatial resolution of two-photon STED microscopy in brain tissue imagingPLOS ONE

Dear Dr. Nemoto,

Thank you for submitting your manuscript to PLOS ONE. After careful consideration, we feel that it has merit but does not fully meet PLOS ONE’s publication criteria as it currently stands. Therefore, we invite you to submit a revised version of the manuscript that addresses the points raised during the review process.

We look forward to receiving your revised manuscript.

Kind regards,

Uma Maheswari Rajagopalan, Ph.D

Academic Editor

PLOS ONE

Journal Requirements:

“This study was supported by the AMED Brain/MINDS JP19dm0207078; by the JST CREST JPMJCR20E4; by the MEXT/JSPS KAKENHI (JP15H05953 “Resonance Bio,” JP16H06280 “Advanced Bioimaging Support,” JP18K14659, JP20H00523, JP20H05669, JP21K19346, JP22H02756); by the MEXT “Network Joint Research Center for Materials and Devices”; by the MEXT “Dynamic Alliance for Open Innovation Bridging Human, Environment and Materials.”; by the Research Foundation for Opto-Science and Technology; and by the ExCELLS “Encouragement Research for Young Scientists”.”

Reviewers' comments:

Reviewer's Responses to Questions

**Comments to the Author**

1. Is the manuscript technically sound, and do the data support the conclusions?

Reviewer #1: Yes

Reviewer #2: Yes

Reviewer #3: Partly

2. Has the statistical analysis been performed appropriately and rigorously? 

Reviewer #1: Yes

Reviewer #2: Yes

Reviewer #3: Yes

3. Have the authors made all data underlying the findings in their manuscript fully available?

Reviewer #1: Yes

Reviewer #2: Yes

Reviewer #3: No

4. Is the manuscript presented in an intelligible fashion and written in standard English?

Reviewer #1: Yes

Reviewer #2: Yes

Reviewer #3: No

5. Review Comments to the Author

Reviewer #1: 1)How to generate and adjust the time delay between 2PE and STED pulses and what is the specific value? Please describe it in detail in the article.

2)The pictures in the paper are suggested to be provided in high resolution.

3)What are the factors that affect the gate window? And how to determine the gate window, including the Tfinish and Tinitial.

4)It is suggested to use signal-to-noise ratio (SNR) or signal-to-back ratio (SBR) to quantify the effectiveness of this method in improving image quality.

Reviewer #2: The manuscript ‘All-electrically synchronized picosecond pulses and time-gated detection improve the spatial resolution of two-photon STED microscopy in brain tissue imaging’ by Ishii et al. introduces 2-photon excitation STED with pulsed depletion and time gated detection. Even if time-gating is mostly utilised along CW based STED microscopy in the existing literature, it here also improves resolution when combined with sub-nanosecond pulsed depletion.

The authors evaluate the new approach on beads and fixed tissue dendritic spines, and report a 1.4 times improvement in lateral spatial resolution in fixed mouse brain slices.

The manuscript is well written and clear, and the findings appear convincing. I have only a few suggestions for improving it:

It would be valuable if the authors could quantify, or at least indicate in the text, how much of the signal is discarded by the gating, for example by reporting max pixel counts observed with and without gating.

I am in doubt why the authors emphasise the synchronisation is “all-electrical”. What would it be if not electrical in nature? Perhaps the authors can better explain this.

The authors should describe if the lines across beads and spine necks were placed manually or using software, and whether a single pixel wide line or multi pixel wide line was used. Also, was the applied Gaussian function with offset to compensate for background fluorescence?

Line 64: “fewer” should be “less”.

Line 126: the sentence “In contrast, the spatial resolution of 2PE-pSTED images of EYFP-labeled neurons was inferior to that of Nile red bead images of which spatial resolution was below 100 nm (Figure 2).” is unclear, perhaps it can be rephrased.

Line 276 onwards, “pixel size of 414 nm … a pixel size of 207 nm…” What is meant by these pixel sizes? Please indicate pixel size as x,y values, e.g. 30 nm by 30 nm, or something similar.

Reviewer #3: Ishii et al. present an upgrade to their two-photon STED microscopy setup, with the main improvement being the introduction of gated detection. This feature is rather standard on single-photon STED instruments (at least on the commercial ones I had the chance to work with), so I was surprised to learn that this has not yet been implemented in two-photon setups. Anyway, it is good to have the effect robustly quantified. However, the current version of the manuscript requires considerable improvements, after which I am happy to review the manuscript again.

Major comments:

1. Figure 2:

a. Why are total photon counts more or less the same for both powers of the STED laser?

b. Resolution improvement by gating looks stronger in B than quantified in C; if B is representative, fitting of a 3D gaussian to I(x,y) may give you more robust results

2. Figure 3 only shows one measurement per condition, so lacks statistics to sufficiently quantify the effect. At least a few more datapoints should be extracted from the same image, even better if from different recordings. If not that important and only used as an illustration, it should be moved to the Supplement and the text modified to clearly convey this lack.

3. Figure 4D: Line 1 seems to contain two peaks that blend into one for confocal, which could explain the outliers in E.

4. Why are the gating windows – seemingly arbitrarily – varied between conditions? Please explain for your case, and give useful guidelines for the reader how to go about it in their case: how to consider parameters like probe lifetime and saturation power, duration of excitation and STED pulses, amount of scattering in their sample, etc.

5. Line 178: “This effectiveness … may be due to the removal of the background noises such as optical scattering from the brain tissues and autofluorescence inherent in biomaterials” – please provide TCSPC histograms from such background/autofluorescence area (i.e. next to dendrites), which should also support your claim in lines 189–192. Measurements at more than two depths could also help.

6. Several claims need further explanation, or reference to specific results or literature:

a. Ad line 53: why is a more “compact” microscope needed?

b. Ad line 57: how does “electrically controllable components” compare to FPGA-based instruments?

c. Ad line 71: what you refer to as “noise” has nothing to do with random fluctuations (statistical/shot noise, read/electronic noise), and should rather be called “background signal” (please correct throughout)

d. Ad line 143: Where did the numbers “1.3 and 3.0 (× 10**3 photons/min/μm2)” come from?

e. Ad line 158: “The shadows must be derived from intracellular organelles … Such shadow imaging has been proposed … ” – citations missing

f. Ad line 173: What was the reason for the low pulse repetition rate in this study?

7. As your setup incorporates the TCSPC detection unit, one could also perform the SPLIT analysis (https://doi.org/10.1039/C8NR07485B). Please discuss both approaches.

8. As per PLOS data policy, I understand that raw images should also be made available.

Minor comments:

9. Would be easier to read if abbreviations (“2PE-gated-pSTED”) are simplified. At the very least, the “p” for pulsed can be omitted, as it is never compared to the non-pulsed version. In text, abbreviations can be replaced with description that is relevant for the comparison “non-gated/gated”

10. Figure 2:

a. the main histogram in A should show the interesting part of the signal, i.e. -10–30 ns

b. check the scalebar in B

11. Figure 4 lacks labels to panels A, B

12. Several parts of text should be made clearer:

a. Ad line 101: “power … was 3.0 and 3.0 mW, respectively” reads as an error, replace by “were both 3.0 mW”

b. Ad line 110: “1.1 times improved” can be dubious – replace with “improved by 10%” or “improved by a factor of 1.1”

13. The manuscript should go through meticulous proof-reading for typos, un-naturally sounding sentences, and grammatical errors, e.g.

a. line 36: “such as superior-penetration depth”

b. line 39: “technologies enabling a spatial resolution at several 10-nanometer or super-resolution imaging”

c. line 43: “several proposals for the improvement of the spatial resolution of 2PE microscopy have been in progress by

combining STED technologies”

d. line 53: “More compact … super-resolution microscopy or 2PE-STED microscopy is required”

e. line 64: “results in fewer photodamages”

f. line 67: “undesirable noises … remains”

g. line 74: “the background noise-derived optical scattering”

h. line 124: “the gate window (ΔT) were set”

…

6. PLOS authors have the option to publish the peer review history of their article (what does this mean?). If published, this will include your full peer review and any attached files.

Reviewer #1: No

Reviewer #2: **Yes: **Jan Tonnesen

Reviewer #3: **Yes: **Iztok Urbancic

---

## [Author Response · Author response to Decision Letter 0]

1 Aug 2023

We would like to thank the reviewers for their careful and insightful comments, which have enabled us to improve the quality of our manuscript. Here are our responses to the reviewer’s comments.

Comments from Reviewer 1

1)How to generate and adjust the time delay between 2PE and STED pulses and what is the specific value? Please describe it in detail in the article.

Author’s response：

We appreciate the helpful comments of Reviewer 1. We have revised Materials and methods to describe how to adjust the time delay in detail. We also applicate to the reviewer’s comment because this comment made us realize an error in Fig 1. The 2PE and STED beam paths against a dichroic mirror were incorrect. We apologize for the error and have corrected the illustration of the optical setup.

Line 303-311 now reads; The laser beams were directed through a single optical pass using a dichroic mirror (FF775-Di01; Semrock, NY, US), and the relative delay timing between the 2PE and STED pulses was measured using a photodetector (1414; New Focus, CA, US) and a sampling oscilloscope (TDS8200, Tektronix, OR, US). Then, an electrical timing controller (T560; Highland Technology, CA, US) was used to add delay to the 2PE pulse at a 10 ps resolution to make it overlap the initial rise point of the STED pulse. In a past study, we confirmed that the relative position of each pulse gave the best STED efficiency [16]. Once the setting of the T560 was determined, the pulses could be reproducibly synchronized when the light sources were turned on. The pulse synchronization did not have to be checked using the oscilloscope each time.

2)The pictures in the paper are suggested to be provided in high resolution.

Author’s response：

We thank Reviewer 1 for this comment. We have uploaded higher-resolution versions of the figures. We have also uploaded all of the raw images used in the figures as S1 Raw images (ZIP file).

3)What are the factors that affect the gate window? And how to determine the gate window, including the Tfinish and Tinitial.

Author’s response：

We thank the precise questions from Reviewer 1. The time-gated detection system was achieved by discarding early arriving photons due to incomplete depletion noise by STED, which negatively affects the spatial resolution. Vicidomini et al. (PLOS ONE, 2013) theoretically revealed that collecting the photons immediately after the STED action produces the sharpest effective point-spread-function for the pulsed STED implementation. The use of a time delay from the excitation pulse (=Tinitial as defined in this study) larger than the STED pulse width only reduces the brightness without further reducing the FWHM. It was also expected that late detection before the next excitation pulse would be dominated by background signal, which also negatively affects the spatial resolution.

In this study, we searched for the gate window (Tinitial and Tfinish) that gave the best FWHM values for the images of beads and spine necks. For the bead images (Fig 2), the FWHM value was evaluated for each gate window for the sum of all 41 images of beads, as shown in S1 Fig. The gate window could be set by the adjustable delay on each time bin width (250 ps). Because the STED pulse width was 260 ps (Fig 1), we expected the best FWHM values to be obtained at Tinitial = 0.25 or 0.50 ns. Tfinish was set arbitrarily. For the spine neck images (Figs 3 and 4), the FWHM value was evaluated for each gate window for one of the necks from the images. Similar to the case of bead images, we searched for the gate windows that gave the best FWHM values, as shown in S3 and S5 Figs. Additionally, we also evaluated the case of Tinitial = 1.25 ns because we expected the autofluorescence inherent in biomaterials, which have a shorter fluorescence lifetime than the target fluorophore. This was a trade-off between the reduction of autofluorescence inherent and the reduction of the brightness, however, we estimated this effect experimentally. Based on these results, we determined the gate window to obtain the best FWHM. We have added S1, S3, and S5 Figs and the figure legends to show the data how to determine the gate windows. We have also revised Results to describe the data.

Line 107-112 now reads; Then, we searched for the gate window ΔT (= Tfinish − Tinitial) that gave the best FWHM values in the bead images (S1 Fig). Vicidomini et al. [21] theoretically revealed that the sharpest effective point spread function is obtained when photons are collected immediately after STED in pulsed STED microscopy. Late detection before the next excitation pulse will lead to the dominance of background signals. Thus, we set Tinitial to 0.25 ns or 0.50 ns (time-bin width = 0.25 ns, STED pulse width = 260 ps) and Tfinish was set arbitrarily (S1 Fig).

Line 149-155 now reads; Then, the FWHM was evaluated at Tinitial values of 0.25 ns, 0.50 ns, and 1.25 ns and arbitrary Tfinish values. Unlike in the case of the bead images (S1 Fig), we also evaluated the FWHM at Tinitial = 1.25 ns because we expected the autofluorescence inherent in biomaterials, which has a shorter lifetime than that of the target fluorophore (S3 Fig). Theoretical results have demonstrated that the use of an initial time (Tinitial) larger than the STED pulse width reduces the brightness without further reducing the FWHM, and we experimentally evaluated the time-gating effects on brain tissues for the first time.

4)It is suggested to use signal-to-noise ratio (SNR) or signal-to-back ratio (SBR) to quantify the effectiveness of this method in improving image quality.

Author’s response：

We thank Reviewer 1 for this comment and agree with this suggestion. In response, we have conducted an analysis of the peak signal-to-background ratio (PSBR), which was utilized by Vicidomini et al. (PLOS ONE, 2013), for all images to assess the quality of the images. We have added the data to Figs 2 to 4 and revised Results and Materials and methods to describe the data.

Line 117-120 now reads; The effectiveness of time-gating was also assessed by comparing the peak signal-to-background ratios (PSBRs) of the time-gated and non-time-gated images (Fig 2E). Time-gating decreased the fluorescence peak and mean background intensities (S2 Fig) and increased the PSBRs of all images, including the 2PE images.

Line 159-160 now reads; Both the fluorescence peaks and background intensities were decreased by time-gating (S4 Fig), but the PSBRs did not change significantly.

Line 196-198 now reads; Although the fluorescence peak and background intensities were decreased by time-gating (S6 Fig), the PSBR in the gated 2PE-STED image was larger than that in the non-gated image (Fig 4F).

Line 378-386 now reads; We used Fiji to measure the peak intensity and mean background intensity of each bead image. The region of interest (ROI) was manually drawn outside the bead as the background region in the 2PE image. The same ROI was placed on all bead images, and each mean intensity was measured as the mean background intensity. The fluorescence peak intensities were calculated by subtracting the background intensities from the raw peak intensities. Then, the PSBR was obtained by dividing the fluorescence peak intensity by the mean background intensity. We used the parameters obtained from the Gaussian curves to calculate the PSBR in the local area around the spine neck. The amplitude and offset were defined as the fluorescence peak intensity and background intensity, respectively. The PSBR for the spine neck was obtained by dividing the amplitude by the offset.

Comments from Reviewer 2

It would be valuable if the authors could quantify, or at least indicate in the text, how much of the signal is discarded by the gating, for example by reporting max pixel counts observed with and without gating.

Author’s response：

We thank the precise comments from Reviewer 1. We have added new analysis data to S2, S4, S6 Figs regarding how much of the signal was discarded by the gating. Additionally, we modified the fluorescence intensity profiles of each bead and spine neck to show the counts of fluorescence photons, in order to more clearly show how much the signal changed by the gating as shown in Figs 3 and 4. 

I am in doubt why the authors emphasise the synchronisation is “all-electrical”. What would it be if not electrical in nature? Perhaps the authors can better explain this.

Author’s response：

Thank you for providing this insight. Our microscope system utilized laser-diode (LD) based pulsed light sources for two-photon excitation (2PE) and STED. The 2PE and the STED pulses could be tightly synchronized using an electrical timing controller, reproducibly. We have revised Materials and methods to describe how to synchronize the pulses in more detail. On the other hand, there were other ways to synchronize pulses without using electronics. For example, our electrical timing controller and optical delay line are thought to be the same in nature for pulse synchronization. As Reviewer 2 pointed out, we emphasized “all-electrical” too much. So, we revised the sentences regarding “all-electrical” including the title of this paper.

Line 303-311 now reads; The laser beams were directed through a single optical pass using a dichroic mirror (FF775-Di01; Semrock, NY, US), and the relative delay timing between the 2PE and STED pulses was measured using a photodetector (1414; New Focus, CA, US) and a sampling oscilloscope (TDS8200, Tektronix, OR, US). Then, an electrical timing controller (T560; Highland Technology, CA, US) was used to add delay to the 2PE pulse at a 10 ps resolution to make it overlap the initial rise point of the STED pulse. In a past study, we confirmed that the relative position of each pulse gave the best STED efficiency [16]. Once the setting of the T560 was determined, the pulses could be reproducibly synchronized when the light sources were turned on. The pulse synchronization did not have to be checked using the oscilloscope each time.

The authors should describe if the lines across beads and spine necks were placed manually or using software, and whether a single pixel wide line or multi pixel wide line was used. Also, was the applied Gaussian function with offset to compensate for background fluorescence?

Author’s response：

We thank Reviewer 2 for this comment and apologize for our insufficient description. We have placed the single pixel wide lines across beads and spine necks manually using ImageJ. Then, the line profiles were fitted with a Gaussian function, GaussAmp in Origin software, to find the parameters of the amplitude A, offset y0, width w, and center xc as follows:

y=y_0+Ae^(-(x-x_c )^2/(2w^2 ))

The full width half maximum (FWHM) of the Gaussian function was calculated using the width parameter:

FWHM = 2w√(ln(4) )

As suggested by the reviewer, we used the offset parameters to compensate for background fluorescence. We have revised Materials and methods to describe in detail how to place the lines and apply the Gaussian function.

Line 370-377 now reads; All fluorescence intensity profiles were obtained using single-pixel-wide lines in the software Fiji. For the beads, the fluorescence intensity profiles across the central intensity were obtained along the x and y axes. For the spine necks, the lines were placed across the spine necks manually. The line profiles were fitted with a Gaussian function (GaussAmp in the software Origin) to find the parameters of the amplitude A, offset y0, width w, and center xc as follows:

y=y_0+Ae^(-(x-x_c )^2/(2w^2 )).

The FWHM of the Gaussian function was calculated using the width parameter:

FWHM = 2w√(ln(4) ).

Line 64: “fewer” should be “less”.

Author’s response：

Thank you for pointing this out. We have corrected this word (Line 62).

Line 126: the sentence “In contrast, the spatial resolution of 2PE-pSTED images of EYFP-labeled neurons was inferior to that of Nile red bead images of which spatial resolution was below 100 nm (Figure 2).” is unclear, perhaps it can be rephrased.

Author’s response：

We thank kind comment from Reviewer 1. We have rephased this sentence.

Line 158-159 now reads; These FWHM values were inferior to those in the 2PE-STED images of the Nile red beads (Fig 2). 

Line 276 onwards, “pixel size of 414 nm … a pixel size of 207 nm…” What is meant by these pixel sizes? Please indicate pixel size as x,y values, e.g. 30 nm by 30 nm, or something similar.

Author’s response：

Thank you for pointing this out. We have revised Materials and methods to express the pixel sizes precisely as Reviewer 2 suggested.

Comments from Reviewer 3

Major comments:

1. Figure 2:

a. Why are total photon counts more or less the same for both powers of the STED laser?

Author’s response：

We thank Reviewer 3 for this comment. It might be probably caused by the photobleaching. We first imaged the Nile red bead at a STED power of 3 mW. Then, we changed the STED power to 1 mW and image the same bead again. We reduced photobleaching by using pulsed STED light, but it was impossible to suppress it completely. We have added a sentence describing this to Discussion.

Line 239-243 now reads; Photobleaching could not be eliminated completely. This was evident in bead imaging, where the total number of photons in the 2PE-STED image at PSTED = 1.0 mW was almost the same as that in the 2PE-STED image at PSTED = 3.0 mW, which was captured first (Fig 2A). This result was probably caused by photobleaching, as the total number of photons should decrease as the STED power increases.

b. Resolution improvement by gating looks stronger in B than quantified in C; if B is representative, fitting of a 3D gaussian to I(x,y) may give you more robust results

Author’s response：

We thank precise comment from Reviewer 3. We agree that we might obtain more robust results if we used 3D gaussian fitting. However, even though the 2D gaussian fitting along x-axis and y-axis was used, we could the FWHM values enough to confirm whether the time-gating was efficient to decrease the FWHM values or not. Also, because the 2PE beam was elliptically polarized by tLCDs, the focal spot was elongated along the polarization direction (= the y-axis in our setup) as described in Results. To show this feature clearly, we believe it was valuable to show the FWHM value along x- and y-axes for this paper.

Line 105-107 now reads; Because the 2PE beam was elliptically polarized using the tLCDs (Materials and methods), the focal spot was elongated along the polarization direction (=the y-axis in our setup).

2. Figure 3 only shows one measurement per condition, so lacks statistics to sufficiently quantify the effect. At least a few more datapoints should be extracted from the same image, even better if from different recordings. If not that important and only used as an illustration, it should be moved to the Supplement and the text modified to clearly convey this lack.

Author’s response：

According to the reviewer’s comments, we have extracted data points from the same image and statistically analyzed the FWHM values per condition. We have revised Fig 3 to more clearly emphasize the effects of time-gating depending on the imaging depth in brain tissues. We have modified a part of Results to explain the data. 

Line 155-158 now reads; The best FWHM values were obtained at Tinitial = 1.25 ns in the images at the 4 µm and 36 µm depths. Setting ΔT to 1.25–14.00 ns for the 2PE-STED image at the 4 µm depth gave an FWHM value of 182.1 ± 10.8 nm, whereas setting ΔT to 1.25–7.00 ns for the 2PE-STED image at the 36 µm depth gave an FWHM value of 178.3 ± 9.4 nm.

3. Figure 4D: Line 1 seems to contain two peaks that blend into one for confocal, which could explain the outliers in E.

Author’s response：

We thank kind comment from Reviewer 3. As indicated, we have added the sentences to explain the outliers in Fig 4E.

Line 194-196 now reads; The line 1 profile in the 2PE image seemingly contained two peaks that blended into one (Fig 4D), which could explain the outliers in Fig 4E. The peak was resolved into two peaks in the 2PE-gSTED image, and one of them reached an FWHM value of 90 nm (Fig 4C).

4. Why are the gating windows – seemingly arbitrarily – varied between conditions? Please explain for your case, and give useful guidelines for the reader how to go about it in their case: how to consider parameters like probe lifetime and saturation power, duration of excitation and STED pulses, amount of scattering in their sample, etc.

Author’s response：

Thank you for providing precise comments. The time-gated detection system was achieved by discarding early arriving photons due to incomplete depletion noise by STED, which negatively affects the spatial resolution. Vicidomini et al. (PLOS ONE, 2013) theoretically revealed that collecting the photons immediately after the STED action produces the sharpest effective point-spread-function for the pulsed STED implementation. The use of a time delay from the excitation pulse (=Tinitial as defined in this study) larger than the STED pulse width only reduces the brightness without further reducing the FWHM. The FWHM is independent of the fluorescence lifetime. It was also expected that late detection before the next excitation pulse would be dominated by background signal, which also negatively affects the spatial resolution. 

In this study, we searched for the gate window (Tinitial and Tfinish) that gave the best FWHM values for the images of beads and spine necks. For the bead images (Fig 2), the FWHM value was evaluated for each gate window for the sum of all 41 images of beads, as shown in S1 Fig. The gate window could be set by the adjustable delay on each time bin width (250 ps). Because the STED pulse width was 260 ps (Fig 1), we expected the best FWHM values to be obtained at Tinitial = 0.25 or 0.50 ns. Tfinish was set arbitrarily. For the spine neck images (Figs 3 and 4), the FWHM value was evaluated for each gate window for one of the necks from the images. Similar to the case of bead images, we searched for the gate windows that gave the best FWHM values, as shown in S3 and S5 Figs. Additionally, we also evaluated the case of Tinitial = 1.25 ns because we expected the autofluorescence inherent in biomaterials, which have a shorter fluorescence lifetime than the target fluorophore. In addition, the scattering photons in the brain tissues were also expected to be excluded more by the delay of Tinitial. But the delay of Tinitia was a trade-off between the reduction of autofluorescence inherent and the reduction of the brightness, however, we estimated this effect experimentally. Based on these results, we determined the gate window to obtain the best FWHM. We have also added S1, S3, and S5 Figs and the figure legends to show the data how to determine the gate windows. We have also revised Results to describe the data.

Line 107-112 now reads; Then, we searched for the gate window ΔT (= Tfinish − Tinitial) that gave the best FWHM values in the bead images (S1 Fig). Vicidomini et al. [21] theoretically revealed that the sharpest effective point spread function is obtained when photons are collected immediately after STED in pulsed STED microscopy. Late detection before the next excitation pulse will lead to the dominance of background signals. Thus, we set Tinitial to 0.25 ns or 0.50 ns (time-bin width = 0.25 ns, STED pulse width = 260 ps) and Tfinish was set arbitrarily (S1 Fig).

Line 149-155 now reads; Then, the FWHM was evaluated at Tinitial values of 0.25 ns, 0.50 ns, and 1.25 ns and arbitrary Tfinish values. Unlike in the case of the bead images (S1 Fig), we also evaluated the FWHM at Tinitial = 1.25 ns because we expected the autofluorescence inherent in biomaterials, which has a shorter lifetime than that of the target fluorophore (S3 Fig). Theoretical results have demonstrated that the use of an initial time (Tinitial) larger than the STED pulse width reduces the brightness without further reducing the FWHM, and we experimentally evaluated the time-gating effects on brain tissues for the first time.

5. Line 178: “This effectiveness … may be due to the removal of the background noises such as optical scattering from the brain tissues and autofluorescence inherent in biomaterials” – please provide TCSPC histograms from such background/autofluorescence area (i.e. next to dendrites), which should also support your claim in lines 189–192. Measurements at more than two depths could also help.

Author’s response：

We thank Reviewer 3 for the valuable suggestions. We set ROI in the background region and created TCSPC histogram from the ROI in the images at 4 µm depth and 36 µm depth (S3 Fig). Unfortunately, it was hard to take the same quality of images from the fixed samples of brain slices that had been prepared for Fig 3. We could not carry out the measurements at more than two depths to compare background signals depending on the imaging depths. However, as the reviewer pointed out, we could clearly show the difference in the background signal related to the fluorescence signal between the images obtained at 4 µm and 36 µm depths. We added the data to S3 Fig and revised Results and Discussion to explain the data, which makes our claim clearer.

Line 150-152 now reads; Unlike in the case of the bead images (S1 Fig), we also evaluated the FWHM at Tinitial = 1.25 ns because we expected the autofluorescence inherent in biomaterials, which has a shorter lifetime than that of the target fluorophore (S3 Fig).

Line 267-268 now reads; The background signal relative to the fluorescence signal was higher in the 2PE-STED image at the 36 µm depth than at the 4 µm depth (S3 Figs A and D).

6. Several claims need further explanation, or reference to specific results or literature:

a. Ad line 53: why is a more “compact” microscope needed?

Author’s response：

Thank you for providing this insight. Large, complex optical setups often require extensive optical alignments that lead to long delays in the observation of biological specimens. We would like to emphasize that it is important to develop a two-photon STED system with a “compact” that is easy-to-use, even for biologists who are not experts in optics. In addition, when considering a system that can be easily installed in many laboratories in the future, the concept of the optical setup itself must be made as compact as possible to save space. However, the main focus of this study is the improvement of spatial resolution by time-gating. We apologize for overemphasizing the compactness of the system, and we have revised the manuscript to reduce the emphasis on this aspect. 

The sentence (Line 53-54) you pointed out now reads; An easy-to-use 2PE-STED microscopy method is required to accelerate the nanoscale visualization of brain functions in many laboratories.

b. Ad line 57: how does “electrically controllable components” compare to FPGA-based instruments?

Author’s response：

We thank Reviewer 2 for this comment. The electrically controllable components” referred to in the manuscript are the laser-diode-based pulsed light source system and the transmissive liquid-crystal devices. The parameters of these components can be electrically controlled, so we expressed them as "electrically controllable". However, we were not referring to integrated circuits or programming components including FPGA, as pointed out by Reviewer 3. We have removed misleading expressions such as "all-electrical" from the manuscript as much as possible.

c. Ad line 71: what you refer to as “noise” has nothing to do with random fluctuations (statistical/shot noise, read/electronic noise), and should rather be called “background signal” (please correct throughout)

Author’s response：

We thank precise comment from Reviewer 3. As Reviewer 3 indicated, “noise” was not an accurate expression. We have replaced “noise” to “background signal” throughout the manuscript. 

d. Ad line 143: Where did the numbers “1.3 and 3.0 (× 10**3 photons/min/μm2)” come from?

Author’s response：

Thank you for pointing it out. We set a region of interest (ROI) around the neuronal dendrite as large as possible and counted the number of photons. We then converted the number of photons to photons per minute and per µm2, to compare the results of 2PE-STED images with and without time-gating. However, as the reviewer indicated, the meaning of this number was not clear. We came to the conclusion that another analysis was appropriate to show that the background signal decreased close to the dendrite after time-gating. We defined the offset parameter obtained from the results of Gaussian fitting as the background signal and compared the value with and without time-gating (Fig S6). The same analysis was carried out for the results in Fig 3 (S4 Fig). 

e. Ad line 158: “The shadows must be derived from intracellular organelles … Such shadow imaging has been proposed … ” – citations missing

Author’s response：

Thank you for pointing out the missing citation. We have added the citation for the article by Tønnesen et al. (Cell, 2018) [23].

f. Ad line 173: What was the reason for the low pulse repetition rate in this study?

Author’s response：

We thank the precise comment from Reviewer 3. We operated the system at a repetition rate of 5 MHz to achieve a high-pulse energy for the STED pulses. For STED, we employed a custom-built laser-diode (LD) based pulsed light source (Hung et al., 2017). At a repetition rate of 5 MHz, the STED pulse had an average power of 7 mW and pulse energy of 1.4 nJ after the output. We had assessed the fluorescence depletion efficiency of the STED pulses against Nile red (~95%), AlexaFluor532 (~90%), and EYFP (~70%) used in this study. We could increase the repetition rate, but the pulse energy would be decreased. Thus, the STED pulses cannot provide efficient depletion at a higher repetition rate. We have added a sentence to explain this point in Discussion. 

Line 248-250 now reads; The repetition rate of our light sources can be increased, but doing so will reduce the pulse energy. Thus, the STED pulses of the current system cannot provide efficient depletion at a higher repetition rate.

7. As your setup incorporates the TCSPC detection unit, one could also perform the SPLIT analysis ( https://doi.org/10.1039/C8NR07485B). Please discuss both approaches.

Author’s response：

We thank Reviewer 3 for the valuable suggestions. We have cited the article and the sentences about the SPLIT analysis approach in Discussion.

Line 278-287 now reads; However, time-gating also rejects photons from the center of the focal spot, resulting in brightness reduction, which might suppress the abovementioned resolution improvement. This problem has been solved through the separation of photons by lifetime tuning (SPLIT) method [25]. This analysis approach uses phasor analysis to efficiently distinguish photons emitted from the center and the periphery of the excitation spot. The spatial resolution of brain tissue imaging using our microscope may be enhanced further via SPLIT analsis.

In conclusion, this study demonstrated that time-gating is more useful for improving spatial resolution in thick brain tissue. Combined with the advantage of the time-gating or other photon separation analyses such as SPLIT, all-pulsed 2PE-gSTED microscopy is expected to facilitate a deeper super-resolution observation to shed light on the brain functions at the nanoscale.

8. As per PLOS data policy, I understand that raw images should also be made available.

Author’s response：

We thank Reviewer 3 for this comment and agree with this suggestion. We have uploaded all of the raw images used in the figures as S1 Raw images (ZIP file).

Minor comments:

9. Would be easier to read if abbreviations (“2PE-gated-pSTED”) are simplified. At the very least, the “p” for pulsed can be omitted, as it is never compared to the non-pulsed version. In text, abbreviations can be replaced with description that is relevant for the comparison “non-gated/gated”

Author’s response：

Thank you for your kind suggestion. We have simplified the abbreviation “2PE-gated-pSTED” to “2PE-gSTED”. We have also replaced the abbreviations in the text with a description of a “non-gated/gated” comparison as much as possible.

10. Figure 2:

a. the main histogram in A should show the interesting part of the signal, i.e. -10–30 ns

Author’s response：

We thank precise comment from Reviewer 3. We have revised the histogram in Fig 2A to show the signal up to 30 ns as suggested.

b. check the scalebar in B

Author’s response：

Thank you for pointing this out. The scalebar was exactly 200 nm. We apologize and make the correction.

11. Figure 4 lacks labels to panels A, B

Author’s response：

Thank you for pointing out the missing labels in Fig 4. We have uploaded a revised version with the labels corrected. 

12. Several parts of text should be made clearer:

a. Ad line 101: “power … was 3.0 and 3.0 mW, respectively” reads as an error, replace by “were both 3.0 mW”

Authors’ response:

Thank you for pointing this out. As Reviewer 3 indicated, the sentence has been corrected.

Line 100-101 now reads; The average power of the 655-nm STED and 2PE beams at the focal plane were both 3.0 mW, respectively.

b. Ad line 110: “1.1 times improved” can be dubious – replace with “improved by 10%” or “improved by a factor of 1.1”

Authors’ response:

Thank you for your kind note. As Reviewer 3 indicated, the sentence has been corrected to “improved by a factor of 1.1”.

Line 114-115 now reads; They were improved by a factor of 1.1 compared to the 2PE-STED images without time-gating.

13. The manuscript should go through meticulous proof-reading for typos, un-naturally sounding sentences, and grammatical errors, e.g.

Author’s response：

We greatly appreciate Reviewer 3 for pointing out grammatical errors in our manuscript. We reviewed all the sentences the reviewer gave us. We also asked English native speakers for further English grammar editing for our manuscript via a proofreading service of Enago (www.enago.jp).

a. line 36: “such as superior-penetration depth”

Author’s response：

The sentence has been corrected.

Line 36-38 now reads; The use of near-infrared pulsed laser beams for 2PE offers superior penetration depths and low invasiveness for biological specimens [2,3].

b. line 39: “technologies enabling a spatial resolution at several 10-nanometer or super-resolution imaging”

Author’s response：

The sentence has been corrected.

Line 38-40 now reads; However, the visualization of nanoscale phenomena, such as morphological changes in neuronal dendritic spines associated with synaptic plasticity, requires super-resolution technologies.

c. line 43: “several proposals for the improvement of the spatial resolution of 2PE microscopy have been in progress by combining STED technologies”

Author’s response：

The sentence has been corrected.

Line 43-44 now reads; Over the past decade, the spatial resolution of 2PE microscopy has been improved by combining STED technologies [5-12].

d. line 53: “More compact … super-resolution microscopy or 2PE-STED microscopy is required”

Author’s response：

The sentence has been corrected.

Line 53-54 now reads; An easy-to-use 2PE-STED microscopy method is required to accelerate the nanoscale visualization of brain functions in many laboratories.

e. line 64: “results in fewer photodamages”

Author’s response：

The sentence has been corrected.

Line 61-63 now reads; Utilizing pulsed laser light for STED reduces the required average power and thus causes less photodamage to specimens than continuous-wave laser light sources.

f. line 67: “undesirable noises … remains”

Author’s response：

The sentence has been corrected.

Line 65-68 now reads; However, the practical use of 2PE-STED microscopy is hindered by spatial resolution degradation caused by background signals from residual fluorescence due to incomplete STED, optical scattering, and the autofluorescence of biomaterials due to the direct excitation of the STED beam.

g. line 74: “the background noise-derived optical scattering”

Author’s response：

The sentence has been corrected.

Line 72-74 now reads; Time-gating is also useful for removing background signals caused by optical scattering and biomaterial autofluorescence, even in the case of STED beams with short pulse widths [20,21].

h. line 124: “the gate window (ΔT) were set”

Author’s response：

The sentence has been corrected.

Line 156-158 now reads; Setting ΔT to 1.25–14.00 ns for the 2PE-STED image at the 4 µm depth gave an FWHM value of 182.1 ± 10.8 nm, whereas setting ΔT to 1.25–7.00 ns for the 2PE-STED image at the 36 µm depth gave an FWHM value of 178.3 ± 9.4 nm.

We believe that our paper is now revised rightly based on the reviewer’s opinion.

Sincerely yours,

Prof. Tomomi Nemoto, Ph. D.

Biophotonics Research Group, Exploratory Research Center on Life and Living Systems (ExCELLS), National Institutes of Natural Sciences

Higashiyama 5-1, Myodaiji, Okazaki, Aichi 444-8787, Japan

Phone: +81-564-59-5255

Email: tn@nips.ac.jp

---

## [Editor Report · Decision Letter 1]

10 Aug 2023

All-synchronized picosecond pulses and time-gated detection improve the spatial resolution of two-photon STED microscopy in brain tissue imaging

PONE-D-23-09651R1

Dear Dr. Nemoto,

We’re pleased to inform you that your manuscript has been judged scientifically suitable for publication and will be formally accepted for publication once it meets all outstanding technical requirements.

Kind regards,

Uma Maheswari Rajagopalan, Ph.D

Academic Editor

PLOS ONE
---

## [Editor Report · Acceptance letter]

15 Aug 2023

PONE-D-23-09651R1 

All-synchronized picosecond pulses and time-gated detection improve the spatial resolution of two-photon STED microscopy in brain tissue imaging 

Dear Dr. Nemoto:

I'm pleased to inform you that your manuscript has been deemed suitable for publication in PLOS ONE. Congratulations! Your manuscript is now with our production department. 

Kind regards, 

on behalf of

Dr. Uma Maheswari Rajagopalan 

Academic Editor

PLOS ONE